

# Assumptions on mixing heights influence the quantification of emission sources: A case study for Cyprus

Imke Hüser[1], Hartwig Harder[1], Angelika Heil[1], and Johannes W. Kaiser[1]

[1]Max Planck Institute for Chemistry, Mainz, Germany

*Correspondence to:* Imke Hüser (imke.hueser@mpic.de), Hartwig Harder (hartwig.harder@mpic.de)

**Abstract.** Lagrangian particle dispersion models (LPDMs) in backward mode are widely-used to quantify the impact of transboundary pollution on downwind sites. Most LPDM applications assume mixing of surface emissions in a boundary layer that is constant in height. The height of this mixing layer (ML), however, is subject to strong spatio-temporal variability. Neglecting this variability may introduce substantial errors in the quantification of source contributions. Here, we perform backward trajectory simulations with the FLEXPART model starting at Cyprus to quantify these errors. The simulations calculate the sensitivity to emissions of upwind pollution sources within the ML height. The emission sensitivity is used to quantify source contributions at the receptor and support the interpretation of ground measurements carried out during the CYPHEX campaign in July 2014. It is determined by two interacting factors: the dilution of pollutants within the ML and the number of trajectories impacted by the emissions. In this study, we calculate the emission sensitivity for a constant ML height of $300\,\mathrm{m}$ and a dynamical ML height to compare the resulting differences. The results show that the impact of emission sources is predominantly overestimated by the neglected dilution in expanded daytime ML heights. There is, however, substantial variability in the simulated differences. For shallow marine or nocturnal ML heights, for example, a ML assumed to high may lead to an underestimation of the intensive concentrations. This variability is predominantly caused by the spatio-temporal changes in ML heights and the meteorological conditions that drive the dispersion of the trajectories. In an application example, the impact of CO emissions from hypothetical forest fires is simulated and source contributions are compared for different ML heights. The resulting difference shows that the $300\,\mathrm{m}$ overestimates the total CO contributions from upwind sources by 16%. Thus, it is recommended to generally implement a dynamic mixing layer height parametrization in LPDMs to prevent these errors.

## 1 Introduction

Local air composition is determined by transport processes in the atmosphere as trace substances can be transported over thousands of kilometres by the wind (e.g., Forster et al., 2001). Long-range transport of pollutants can drastically impair air quality in downwind areas and can trigger exceedances of ambient air quality thresholds (e.g., Vardoulakis and Kassomenos, 2008; Poupkou et al., 2014; Lelieveld et al., 2015). Therefore, the identification of upwind emission sources and the quantification of their impact by atmospheric dispersion modelling is crucial for implementing effective air pollution abatement measures.

One of the main tools for analysing the transport pathways of air pollutants from their source regions to a measurement site are trajectory models. These models track the path of an air tracer particle forward or backward in time in order to establish




relationships between the pollution source and the receptor site (Stohl, 1998). Following the recommendation of Stohl et al. (2002), the backward simulation with Lagrangian Particle Dispersion Models (LPDMs) are most suitable for the interpretation of atmospheric trace substances measurements. LPDMs calculate the trajectories of a large number tracer particles in the mean wind and in statistically modelled turbulent fluctuations (Rodean, 1996; Stohl, 1998). This allows for a realistic representation

of the dispersion of an air mass by turbulence and convection in the lower troposphere (Stohl et al., 2002). Hence, LPDMs such as FLEXPART (Stohl et al., 1998; Stohl and Thomson, 1999; Stohl et al., 2005) are frequently used to identify source regions of pollutions and to quantify their contributions to the atmospheric composition at a measurement site (e.g. Stohl et al., 2007; Lal et al., 2014). Furthermore, these models are used in an inverse mode to estimate the source strength of emissions. Here, the emission fluxes are adjusted to optimize the agreement between simulated and observed concentrations (Stohl et al., 2009).

This inverse method is also applied to validate or improve emission inventories of atmospheric species (Pan et al., 2014).

The methodology in LPDMs is based on the assumption that any regions passed by the trajectories can potentially affect the receptor site. Since emissions from pollution sources are distributed over a well-mixed layer adjacent to the ground, only trajectories passing this layer identify potential source regions. Thus, the source receptor relationship is derived from the trajectories' residence time within this well-mixed layer in the upwind area of influence as described in Seibert and Frank

(2004). It is a function of space and time, that maps quantitatively the relative contribution of pollution sources per unit strength of the emission flux to the air composition at the receptor. Therefore, it is also termed 'footprint function', the 'source weight function' (Schmid, 1994; Kljun et al., 2002) or 'emission sensitivity' (Stohl et al., 2007), the term used here. The height of this well-mixed layer generally corresponds to the mixing layer (ML) height of the planetary boundary layer (PBL). This layer is characterized by turbulent processes and uniformly mixes conservative tracers with a time scale of about an hour or less

(Stull, 1988). Typically, applications of source identification with LPDMs use constant ML heights of $100\,\mathrm{m}$ (e.g., Stohl et al., 2007; Van Dam et al., 2013; Pan et al., 2014), $150\,\mathrm{m}$ (Duck et al., 2007) or $300\,\mathrm{m}$ (Lal et al., 2014). This simplified assumption avoids to exceed PBL heights and to involve air masses of the free troposphere. Furthermore, the dilution of surface emission fluxes is constant and therefore the pollution emitted to the ML is constant per unit strength. With this simplification, the pick up of pollutants only depends on the residence time of trajectories within the ML height. The residence time in the ML is

thus used to describe the relationship of the receptor concentration to emission surface flux of inert species (Seibert and Frank, 2004).

However, PBL heights are subject to pronounced spatial and temporal variability and can reach more than 2000 meters. The dynamical behaviour of the PBL height is thus in contrast to the assumption of a constant ML height with a few hundred meter. If the constant ML height assumptions is replaced by such a dynamic ML height, the emission sensitivity is modified by

two counteracting effects. On the one hand, an increase in ML height leads to a stronger dilution of emission fluxes over this layer - an effect that reduces emission sensitivity. On the other hand, a larger number of trajectories passes through a ML with increased height that increases the residence time for the pick up of pollutants - an effect that increases emission sensitivity.

The aim of this study is to quantify the errors associated with the assumption of a constant ML height. We use backward simulations with the LPDM model FLEXPART that were carried during the CYPHEX ground campaign in July 2014 to study

the source areas and transport routes of air pollution at Cyprus.



In section 2 we derive the emission sensitivity function from the simulations and introduce our modifications to calculate the sensitivity for a dynamical ML height. Furthermore, we develop a method to compare the effects of different ML height assumptions. In section 3.1 we compare the emission sensitivity for a constant ML height of $300\,\mathrm{m}$ and a dynamical ML height. In an application example in section 3.2, we calculate the errors in source contributions of CO that result from hypothetical
forest fires. Finally, in section 4 we summarize and discuss the results.

## 2  Method

### 2.1  FLEXPART simulations

We have performed backward simulations with the LPDM model FLEXPART 9.2 (Stohl et al., 2005) during the CYPHEX campaign. The FLEXPART model simulations have been driven by operational meteorological input data of the European
Centre for Medium-Range Weather Forecasts (ECMWF) with $0.2° \times 0.2°$ spatial resolution (derived from T799 spectral truncation). The model domain extends from $20°\mathrm{W}$ to $70°\mathrm{E}$ and $20°\mathrm{N}$ to $70°\mathrm{N}$ covering Europe, northern Africa and western Asia. To provide a temporal resolution of 1 hour a combination of analyses at 00, 06, 12 and 18 UTC and short forecasts data at intermediate time steps was used. Backward simulations were started from the measurement point in the Northwest of Cyprus by releasing 10000 neutral inert air tracer particles during an 1 hour time interval which are followed as trajectories over 5
days.

The positions of the tracer particles are calculated on a 3-d grid with a horizontal resolution of $0.2°$ that corresponds to the resolution of the input data on the model domain. A vertical resolution of 58 layers is used extending from $100\,\mathrm{m}$ (minimum height of the planetary boundary layer set in the model) to an altitude of $10000\,\mathrm{m}$. The height of the layers gradually increases from $20\,\mathrm{m}$ to $1000\,\mathrm{m}$. The temporal resolution corresponds to the input data and is available in hourly time steps within the
simulation period of $120\,\mathrm{h}$. In total 216 simulations covering 9th July 2014 00:00 UTC to 4th August 2014 21:00 UTC in 3 hour time steps were carried out. They cover the entire period of the measurement campaign.

The output of FLEXPART is a 4-dim function of emission sensitivity (Stohl et al., 2007), three space dimension plus time, derived from the positions of all trajectories. It describes the relation between any emission source that is passed by the trajectories and the concentration of the respective atmospheric substance at the receptor. Since atmospheric emissions are
diluted by mixing in adjacent air, emission sources are specified as emitted mass per time and volume in units of $\mathrm{kg\,m^{-3}\,s^{-1}}$, denoted by $q$. Receptor concentrations are expressed in a conservative quantity as mass mixing ratios, specified by $\chi$ (unitless). Then, the emission sensitivity is determined by the sum of the residence time $T$ of all trajectories in this adjacent volume of air divided by the local air density $\rho$ in $\mathrm{kg\,m^{-3}}$ without transmission correction (Seibert and Frank, 2004)

$$\frac{\partial \chi}{\partial q_{ijkn}} = \frac{T_{ijkn}}{\rho_{ijkn}}. \tag{1}$$

It is computed in the predefined 3-d output grid that uses the indices $i, j, k$ to specify the spatial position $x_i, y_j, z_k$ in the centre of each grid box. The fourth index $n$ determines the time step $t_n$ within the simulation period. This emission sensitivity can be





interpreted as a source-weight factor. It describes the source contribution to the mass mixing ratio at the receptor relative to the source strength, when any transport losses are ignored.

In addition to the described standard output, FLEXPART provides the PBL height for all individual tracer particles at their geographical position at each timestep. The PBL heights are calculated by the model according to Vogelezang and Holtslag (1996) and use the concept of the critical Richardson number that is based on surface sensible heat fluxes and surface stresses available from the ECMWF input data (Section 6 in Stohl et al., 2005).

## 2.2 Surface Emission Sensitivity

Since we are interested in area sources at ground level, the emission sensitivity is slightly modified to derive the sensitivity to surface emissions. The emission flux of an area surface source $q_A$ in units of $\mathrm{kg\,s^{-1}\,m^{-2}}$ is diluted over a vertical layer adjacent to the ground, the mixing layer (ML) of height $h$, following $q = q_A\,h^{-1}$. We use different ML heights and have implemented two methods:

- The first method uses the assumption of a constant ML height within the PBL height (Seibert and Frank, 2004). Here, we chose a layer of $h = 300\,\mathrm{m}$ that is close to the typically expected minimum of PBL heights. Then, all grid boxes with height $z_k \leq h$ are attributed to the mixing volume. The emission sensitivity $S$ to surface area sources is obtained from eq. 1 with the density-weighted residence time $\widehat{T}$ in this mixing volume as

$$\widehat{T}_{ijn} = \sum_{k=1}^{z_k \leq h} \frac{T_{ijkn}}{\rho_{ijkn}} \tag{2}$$

$$S_{ijn} = \frac{\widehat{T}_{ijn}}{h}. \tag{3}$$

The emission sensitivity $S$ in units of $\mathrm{s\,m^2\,kg}$ is proportional to the density-weighted residence time of all trajectories within the ML height. Even for spatial or temporal integration, the emission sensitivity is independent of the ML height.

- The second method is new in this study and uses a dynamic ML height. This ML height $h_{ijn}$ is variable in space and time and specified by the corresponding indices $i, j, n$. Then, eq. 2 and 3 are replaced by

$$\widehat{T}_{ijn} = \sum_{k=1}^{z_k \leq h_{ijn}} \frac{T_{ijkn}}{\rho_{ijkn}} \tag{4}$$

$$S_{ijn} = \frac{\widehat{T}_{ijn}}{h_{ijn}}. \tag{5}$$

With this dynamic ML assumption, the emission sensitivity depends on local varying ML heights. It is hence not proportional to the density-weighted residence time, when spatially or temporal integrated.

To implement this dynamic ML height, we use local PBL heights calculated by the FLEXPART model. However, from the output of FLEXPART, only PBL heights along the particles' trajectories are available. To obtain gridded PBL heights





from all single particle positions, we adapt the methodology described in Stohl et al. (2005), section 11.1. Their eq. (55) is modified and calculates the PBL height $h$ on the spatio-temporal output grid

$$h = \sum_{p=1}^{N} (f_p \, h_p) \tag{6}$$

with $N$ being the total number of particles, $h_p$ the PBL height of particle $p$ and $f_p$ the fraction of the particle attributed to the respective grid cell. To calculate this fraction, we use a uniform kernel with bandwidths of $0.2°$ that corresponds to the output grid.

## 2.3 Analysing the impact of mixing layer height assumptions

To analyse the impact of ML height variations, we calculate the emission sensitivity for both ML heights and analyse their differences. The absolute difference is described by

$$\Delta S_{ijn} = S_{ijn}(h_{ijn}) - S_{ijn}(h). \tag{7}$$

Here, $\Delta S_{ijn}$ is defined such that it is positive when ML height differences introduce a higher emission sensitivity compared to the constant $300\,\mathrm{m}$ ML assumption. Differences in ML height are determined in the same way, $\Delta h_{ijn} = h_{ijn} - h$, with positive sign for an increase.

To analyse the changes in emission sensitivity that are introduced by varying ML heights, Eq. 5 is differentiated with respect to $h$,

$$\frac{dS(h)}{dh} = \frac{d}{dh} \frac{\widehat{T}(h)}{h} = \frac{1}{h} \frac{d\widehat{T}}{dh} - \frac{1}{h^2} \widehat{T}. \tag{8}$$

Rearranging the equation to

$$\frac{dS(h)}{dh} = \frac{\widehat{T}}{h} \frac{1}{dh} \left( \frac{d\widehat{T}}{\widehat{T}} - \frac{dh}{h} \right) \tag{9}$$

and discretising the differentials results in the simple relation

$$\frac{\Delta S}{S} = \frac{\Delta \widehat{T}}{\widehat{T}} - \frac{\Delta h}{h}. \tag{10}$$

The relative change in density-weighted residence time $\left| \frac{\Delta \widehat{T}}{\widehat{T}} \right|$ describes the gain / loss in impact, since a deeper ML height can capture more trajectories. The second term describes the relative change in ML height $\left| \frac{\Delta h}{h} \right|$. It quantifies the dilution of emitted substances and represents the gain / loss in concentration. Thus, to describe the relative change in emission sensitivity, we need to analyse the overall difference of both effects. Additionally, both effects are coupled and interact with each other. An increase in ML height results in an increase in residence time that is accompanied by a stronger dilution. Hence, the changes in emission sensitivity are the result of the counteracting effects: gain in impact and loss in concentration (dilution effect). Crucial is which of both turns out to be the dominating effect. This interaction is described in a case-by-case analysis presented in Tab. 1 respectively for increasing and decreasing ML heights.





**Table 1.** Subdivision of the difference in emission sensitivity $\Delta S$ caused by ML height variations $\Delta h$ assigned to their dominating effects.

| cases of interaction | dominating effect | |
| --- | --- | --- |
| $\Delta h > 0$ and $\Delta S < 0$ | $\left|\frac{\Delta \widehat{T}}{\widehat{T}}\right| < \left|\frac{\Delta h}{h}\right|$ | loss in concentration (stronger dilution) |
| $\Delta h > 0$ and $\Delta S > 0$ | $\left|\frac{\Delta \widehat{T}}{\widehat{T}}\right| > \left|\frac{\Delta h}{h}\right|$ | gain in impact |
| $\Delta h < 0$ and $\Delta S > 0$ | $\left|\frac{\Delta \widehat{T}}{\widehat{T}}\right| < \left|\frac{\Delta h}{h}\right|$ | gain in concentration (less dilution) |
| $\Delta h < 0$ and $\Delta S < 0$ | $\left|\frac{\Delta \widehat{T}}{\widehat{T}}\right| > \left|\frac{\Delta h}{h}\right|$ | loss in impact |

Following this case-by-case analysis, the overall difference in emission sensitivity, spatially and / or temporally integrated, is the results of these four characteristic differences

$$\underbrace{\Delta S_{tot}}_{\text{overall difference}} = \underbrace{\Delta S^- (\Delta h^+)}_{\text{dilution effect} <0} + \underbrace{\Delta S^+ (\Delta h^+)}_{\text{gain in impact} >0} + \underbrace{\Delta S^+ (\Delta h^-)}_{\text{gain in concentration} >0} + \underbrace{\Delta S^- (\Delta h^-)}_{\text{loss in impact} <0}. \tag{11}$$

Here, $\Delta S^{\pm}$ respectively $\Delta h^{\pm}$ are defined according to

$$5 \quad \Delta X^+ = \begin{cases} \Delta X & , \Delta X > 0 \\ 0 & , \Delta X < 0 \end{cases} \tag{12}$$

and

$$\Delta X^- = \begin{cases} 0 & , \Delta X > 0 \\ \Delta X & , \Delta X < 0. \end{cases} \tag{13}$$

The contributions interfere with each other and can buffer the overall difference. However, the individual contributions account for the variability that is introduced by changes in ML heights. The negative and positive contributions span up a margin that

10 envelopes the overall difference. Therefore, this margin is a simplified tool to quantify the potential of errors in source area analysis that rely on a constant ML height.

## 2.4 Simulation of CO source contributions

The fields of emission sensitivity $S_{ijn}$ on the spatio-temporal output grid (spatial indices: $i, j$, temporal index: $n$) derived from the FLEXPART simulations have been calculated for inert substances. Therefore, it quantifies the relative source contribution

15 from any emitted species that are conserved on time scales used here, such as for example carbon monoxide. To derive the absolute source contribution from CO emission sources, we need an additional field of CO emission mass flux density on the spatio-temporal grid: $q_{ijn}^{\mathrm{CO}}$ in $\mathrm{kg\,s^{-1}\,m^{-2}}$. Folding the sensitivity with the emission flux density and temporal integration results in a new field of potential CO source contributions

$$\chi_{ij}^{\mathrm{CO}}(t) = \sum_n \left( S_{ijn}(t) \cdot q_{ijn}^{\mathrm{CO}} \right). \tag{14}$$





$\chi_{ij}^{CO}$ is a unit less quantity and describes the CO contribution from the respective grid cell to the measured CO mass mixing ratio at the receptor point at time $t$. Therefore, it is used to identify the geographical distribution of sources and their impact. In general, a strong impact is expected, if enhanced emission sensitivity encounters a strong emission flux. However, if errors in calculated emission sensitivities encounter strong local emission sources, the error in source contributions is expected to

increase.

Spatial integration finally results in the total mass mixing ratio that is transported to the receptor. It represents the CO contribution due to transport process and should be quantitatively comparable to the measured enhancement over the typical background concentration $\chi_0$ (Seibert and Frank, 2004; Stohl et al., 2007). The observed CO mass mixing ratio $\chi_{obs}$ at the receptor at time $t$ is thus the sum of the background mass mixing ratio and the transport contribution

$$\chi_{obs}(t) = \chi_0 + \sum_{i,j}\sum_n \left( S_{ijn}(t) \cdot q_{ijn}^{CO} \right) + \epsilon. \tag{15}$$

The uncertainty $\epsilon$ accounts for older CO contributions prior to the simulation period, for unknown CO sources and model errors such as neglected transport losses.

In this study, we use the Global Fire Assimilation System GFAS1.2 (Kaiser et al., 2012) to quantify the CO source contributions from biomass burning of forest fires to the CO concentration at Cyprus during the CYPHEX campaign. These CO source

contributions are calculated for both, the constant ML height of $300\,\mathrm{m}$ and a dynamic ML height. Furthermore, the differences are compared and analysed as described for the emission sensitivity in section 2.3. However, in the period of the campaign, fire intensity was to weak to influence the CO concentration significantly. To analyse the impact of errors in calculated emission sensitivities, a strong fire event is necessary. Thus, we use a CO emission pattern of a large biomass burning event that occurred in Greece in August 2007 (Poupkou et al., 2014) to represent such a hypothetical strong fire event.

## 3 Results

### 3.1 Differences in emission sensitivity

#### 3.1.1 A single case study

First, we analyse the impact of ML height variations in a case study with a pronounced diurnal cycle in PBL height. Typically, the PBL height peaks during daytime over heated land masses and reaches heights up to $2-3\,\mathrm{km}$. After sunset the layer

collapses down to $100-300\,\mathrm{m}$ or even lower when surfaces cool down. Therefore, when compared to a constant ML height assumption of $300\,\mathrm{m}$, the error to local PBL heights is positive over day and negative over night. These differences in ML height introduce changes in emission sensitivity following Eq. 7 and thereby influence the simulation of upwind source contributions. To analyse this, we chose the simulation of the 19th July 2014 00 UTC as a case study. It describes the transport of air masses from South-eastern Europe to Cyprus on a continental transport route. Thus, a distinct variability in PBL heights can

be expected.





As a starting point, the upwind emission sources are identified using the constant $300\,\mathrm{m}$ ML height. In a first step, the horizontal field of emission sensitivity $S_{ijn}$, following Eq. 3, is integrated in time. Then, the 2-d field is mapped as shown in Fig. 1a and indicates regions of enhanced emission sensitivity by colors. The higher the sensitivity values, the higher the contribution of an area source. Hence, they identify upwind source regions and quantify their impact on the measurement site.

The next step is to analyse the differences in emission sensitivity that result from local varying PBL heights. First, the temporal variability is neglected and the differences $\Delta S_{ijn}$ are integrated in time. The obtained 2-d spatial field is shown on a map in Fig. 1b. It indicates relative differences in emission sensitivity compared to the application of the $300\,\mathrm{m}$ ML height. Thus, it quantifies the relative difference in reference to the local source strength. In this example, the spatial distribution of differences in emission sensitivity exhibits a pattern of red and blue colors. It represents alternating positive and negative

differences next to each other. Therefore, a systematic trend based on any spatial surface properties is not given. In fact, the 'fixed' ML height assumption capitalises on these local maxima and minima compensating each other.

    To further explore the impact of temporal variations in ML height, the emission sensitivity is spatially integrated. Then, it only depends on time and its evolution can be represented in a simple time profile over the simulation period. For a comparison, this time profile is calculated for the emission sensitivity of the constant $300\,\mathrm{m}$ ML height and local PBL heights and shown in

Fig. 2a. The color of the filled areas indicates the sign of the difference and represents a regular alternating pattern. This is the result of the characteristics of a diurnal cycle that leads to a pronounced temporal dependency. During daytime, the changes in PBL height introduce a loss in emission sensitivity and a gain over night. This suggests that stronger dilution in daytime ML heights reduces the emission sensitivity, even though more trajectories are influenced by emissions.

    To further analyse this effect, the density-weighted residence time $\widehat{T}$ (see Eq. 2 and 4) of all trajectories within the ML

height is integrated in time. The time profile is shown in Fig. 2b for the constant $300\,\mathrm{m}$ ML and the varying PBL height with differences identified by colors. The different colors indicate a strong gain in residence time during daytime and a loss at night that correlates with the diurnal cycle of PBL heights. Furthermore, the comparison of both time profiles in Fig. 2 points out that the gain in residence time over day is linked to a loss in emission sensitivity. Consequently, the dilution compensates for the gain in impact and vice versa. Averaged over the whole time period, the loss in emission sensitivity dominates by $-4\,\%$.

However, this mean value covers smaller positive differences that occur at night time.

    To account for the local and temporal variability in more detail, the overall difference is subdivided in contributions of the individual effects following Eq. 11 and Tab. 1. Theoretically, when PBL heights exceed the constant $300\,\mathrm{m}$ ML height assumption, the emission sensitivity can either decrease or increase. In this example, the stronger dilution predominately compensates for the gain in impact over the whole period. This effect contributes by $-8.8\,\%$ to the overall difference in

emission sensitivity. However, in a few cases the dilution effect effect is less dominant and the gain in impact causes a positive contribution of $2.2\,\%$. When the PBL heights fall below $300\,\mathrm{m}$, the described effects are reversed. Then, an increase in emission sensitivity by $6.2\,\%$ is observed due to the intensified concentration in shallow ML heights. However, when the loss in impact is stronger than this concentration effect, the emission sensitivity decreases. This effect contributes by $-3.6\,\%$ to the overall difference.





By merging these four separated effects, a margin between $-12.4\%$ to $+8.4\%$ is spanned. This margin envelopes the mean difference in emission sensitivity of $-4\%$ and represents the effects of local and temporal variability. Therefore, source contributions from emissions can be over and underestimated, when the constant $300\,\mathrm{m}$ ML height assumption is used.

### 3.1.2 Analysing the whole set of case studies

The analysed single case study points out that changes in emission sensitivity result from the competing effects of dilution and gain in residence time. While the dilution only depends on local ML heights, the residence times of captured trajectories additionally depends on the vertical distribution of air tracers. Hence, it is controlled by the meteorological conditions during transport. To account for a variety of synoptic situations, the analysis is extended on the whole available set of 216 case studies. Typically in summertime, a dominant pattern of northerly winds is expected in the Eastern Mediterranean that drives

the transport of continental air masses from Eastern Europe. However, four distinct periods of west and south westerly flow in the Eastern Mediterranean were observed in July 2014 that cut off the expected continental transport routes and brought maritime air to Cyprus (Tyrlis et al., 2015). These alternating wind pattern offer a variety of meteorological conditions.

To compare different case studies, the procedure of the previous section, following Eq. 11, is applied. For each case study, the overall and the margin of individual differences in emission sensitivity is calculated. The results are shown in a time profile

over the period of the CYPHEX campaign in Fig. 3. Here, the individual contributions are specified by different colors and envelop the overall difference as an uncertainty margin. By representing the temporal evolution over the time period of a month, changes on the temporal scale of days and weeks are involved. Therefore, this overview accounts for the impact of different synoptic patterns.

At first sight, it is obvious that negative differences dominate with mean values around $-5\%$. However, the wide margin

points out that the variability allows for differences of $10\%$ and even more in both directions. For a detailed discussion, the differences are calculated as a mean over all 216 simulations combined with the interquartile range. The results are summarized in a box-whisker plot in Fig. 4 and corresponding values of mean values, box and whisker ranges listed in Table 2. The box-whisker plot represents the spreading of data points and can be used to quantify the impact of each effect. Additionally, it accounts for the variability and includes single outliers that occur in changing synoptic conditions.

The detailed analysis of all case studies confirms that the impact of emission sources is predominately overestimated during summer at Cyprus, when a constant $300\,\mathrm{m}$ ML height assumption is used. For most simulations, the differences in ML height introduce an overall mean difference in emission sensitivity between $-2.8$ and $-7.4\%$. This negative effect is mainly driven by the neglected dilution in PBL heights that exceed the $300\,\mathrm{m}$ ML height assumption. For the majority of case studies, the dilution effect dominates the gain in impact and contributes with $6-11\%$ negatively to the mean overall difference. However,

this changes when air tracers agglomerate above the standard $300\,\mathrm{m}$ ML height and thereby are captured by the overlying PBL height. Then, the gain in impact dominates and leads to an increase in emission sensitivity of about $2-3\%$.

When PBL heights fall below $300\,\mathrm{m}$ the concentration of emitted mass flux is intensified and less trajectories captured by the shallower layer. This gain in concentration counteracts the decreasing residence time and contributes by $4-9\%$ to the difference in emission sensitivity. However, this is reversed, when the air tracers agglomerate between $100$ and $300\,\mathrm{m}$





**Table 2.** Mean values and calculated intervals in per cent for the interquartile range and the 2nd to 98th percentile as shown in the box whisker plot in Fig.4.

| effect | mean | 25%-75% box range | 2%-98% whisker range |
|---|---|---|---|
| dilution | $-8.5$ | $[-10.6\,,\,-6.1]$ | $[-14.1\,,\,-2.7]$ |
| gain in impact | $+2.6$ | $[+2.0\,,\,+3.2]$ | $[+1.0\,,\,+4.4]$ |
| loss in impact | $-6.1$ | $[-7.5\,,\,-3.6]$ | $[-19.9\,,\,-1.5]$ |
| gain in concentration | $+6.9$ | $[+4.1\,,\,+8.9]$ | $[+1.5\,,\,+16.7]$ |
| overall | $-5.0$ | $[-7.4\,,\,-2.8]$ | $[-11.8\,,\,+4.2]$ |

height that co-occurs with a shallow PBL height of about $100\,\mathrm{m}$. Such conditions typically occur, when hot continental air masses reach the relatively cooler sea surface and travel slightly above the shallow marine PBL height. Then, the air tracers are captured by the $300\,\mathrm{m}$ ML height assumption, even though they are in the free troposphere. According to this, a lower ML height decreases the emission sensitivity due to a loss in impact. This negative effect contributes about $4-8\,\%$. However, the

strong outliers of up to $-20\,\%$ represent a pronounced variability of this difference. This variability emphasizes the risk of a false identification of emission sources within the constant $300\,\mathrm{m}$ ML height.

Overall, a negative difference in emission sensitivity is observed. However, 21 positive outliers with a maximum of $9\,\%$ were found among the 216 simulations and the 98th percentile represents a positive difference of $+4.2\,\%$. This variability limits a distinct statement regarding the sign of errors that is introduced by neglected ML height variations. Hence, the impact of

emission sources can be over- and underestimated and depends on local PBL heights and meteorological conditions.

## 3.2 Differences in source contributions

### 3.2.1 Application - Simulation of extreme, localised CO source contributions

The analysis in the previous section reveals that the assumption of a constant $300\,\mathrm{m}$ ML height over and underestimates the sensitivity to emission sources. However, the impact of upwind emission sources additionally depends on the local distribution

and source strength of pollutants. To analyse these errors, we simulate the CO source contributions from emissions of forest fires as described in section 2.4. The strong CO emissions in Greece during August 2007 are shown in Fig. 5a. Furthermore, the simulation of 30th July 2014 shows transport routes that pass the fire regions. This is indicated in the corresponding map of temporal integrated emission sensitivities based on the constant $300\,\mathrm{m}$ ML height in Fig. 5b. Hence, a pick up of CO emissions and impact at the receptor can be expected in this hypothetical case.

The spatial distribution of CO source contributions is calculated following eq. 14 and shown in Fig. 5c. In total, the Greek forest fires cause a CO contribution of $58.8\,\mathrm{ppb}$. This would be a significant enhancement on top of the background CO at





Cyprus of approximately $70\,\mathrm{ppb}$ which is the minimal value measured during the CYPHEX campaign (personal correspondence Uwe Parchatka) in unpolluted air.

To analyse the errors that are introduced by ML height variations, the CO contributions are also calculated using varying PBL heights. The absolute differences are integrated in time and the spatial distribution is represented in Fig. 5d by different colors.

Since only negative values are found over Greece, the ML height variations cause an overestimation of the CO concentration at the receptor. This overestimation sums up to a total difference of $-9.6\,\mathrm{ppb}$ or $-16\,\%$.

To further analyse the impact of the diurnal cycle in PBL heights, the 3-d field of CO source contributions is spatially integrated. Then, the total pick up of CO at each time step is calculated and can be shown in a time profile. In Fig. 6a, this pick up of CO contributions is shown for the time window, when the Greek forest fires would have been passed by the trajectories.

It reveals that the major pick-up takes place over day between 9 and 18 local time. This is accompanied by a negative difference of more than $20\,\%$ and hence a pronounced overestimation of CO contributions.

During this time window 12 hours prior at 28th July 12:00 UTC (15:00 local time), the PBL height reaches a mean daytime peak of $1500\,\mathrm{m}$. Therefore, the stronger dilution decreases the concentration of emitted pollutants by $80\,\%$. On the other hand, the residence time of trajectories captured within this layer increases and reinforces the total pick up of emissions.

For a detailed analysis of this interaction, the time profile of corresponding density-weighted residence time $\widehat{T}$ is calculated with eq. 2 and 4. In Fig. 6b, it is arranged next to the time profile of CO contributions. It verifies that the residence time increases by more than twice at time step $-12\,\mathrm{h}$. However, this gain in impact is not sufficient to compensate the strong dilution. The CO contribution from the PBL height is lower than from a constant ML height of $300\,\mathrm{m}$. Therefore, the neglected dilution can be identified as the driving mechanism that leads to an overestimation of emission impact. In addition, this effect is amplified,

since the emission sources are passed in a time window with a pronounced negative difference in emission sensitivity.

## 4 Conclusions

In this study, we have compared the source identification with the LPDM model FLEXPART for two different mixing layer (ML) height assumptions. The assumed ML height determines the dilution of emitted pollutants at ground level and the number of passing trajectories that are exposed to the emissions. Furthermore, both effects counteract with each other: a growth in ML

height leads to a stronger dilution that is coupled to a stronger impact. Therefore, the effects of this ML height assumption on the quantification of upwind emission sources are subject to discussion. Here, we distinguish four different effects. In case of a growing ML height, we analyse the effect of

- loss in concentration (dilution effect)

- gain in impact.

In case of a decreasing ML height, the reverse effects of

- gain in concentration





– loss in impact

are analysed.

First, we used the recommended assumption of a constant ML height of $300\,\mathrm{m}$ that is close to the minimum of the planetary boundary layer (PBL) height (Seibert and Frank, 2004). This simplification assumes a constant dilution and decouples the two effects. Then, the sensitivity to emissions is proportional to the residence time of the trajectory ensemble in the ML and spatio-temporal independent. Additionally, this assumption lowers the risk to falsely consider trajectories outside the ML height. However, it fails to consider stronger dilution and influencing more trajectories in the convective ML during daytime. Then, we compared the source quantification of the $300\,\mathrm{m}$ ML height assumption to that of a spatio-temporal ML height. This variable ML height is realized by the PBL heights that are derived from the model. Since the PBL height follows a pronounced diurnal cycle, it provides a dynamic ML height that depends on local conditions and the time of the day. Hence, it is suited for the purpose here even though the modelled PBL height in FLEXPART is subject to uncertainties (Berkes, 2014).

For the comparison, we used in total 216 FLEXPART simulations that were carried out during the CYPHEX campaign at Cyprus in July 2014. The case studies provide different meteorological conditions and transport patterns. With this variety, differences in emission sensitivity are quantified for different local conditions:

– When the PBL height exceeds the $300\,\mathrm{m}$ height assumption, we observed a decrease in emission sensitivity by $5.9\,\%$ in average. The variability of local conditions allows for a decrease of $10\,\%$ and more. Positive differences of more than $3\,\%$ are rarely observed. Thus, the strong dilution in extended ML heights dominates the accompanied gain in impact. The $300\,\mathrm{m}$ ML assumption overestimates the impact of source regions due to this neglected dilution.

– When the PBL height falls below $300\,\mathrm{m}$, a weak increase of $0.8\,\%$ is observed in average. It results from two counteracting effects. The gain in concentration is compensated by the accompanied loss in impact. Accordingly, the temporal and spatial variability reveals both an increase and a decrease in emission sensitivity of $4-8\,\%$ with outliers of more than $15\,\%$. The dominant effect depends strongly on the vertical distribution of air tracer particles in the lower troposphere. In a homogeneous vertical distribution, the gain in concentration dominates. Hence, the $300\,\mathrm{m}$ height assumption underestimates the impact of emission sources. In reverse, when air tracers agglomerate between the $300\,\mathrm{m}$ and the shallow PBL height below, they are falsely influenced within the $300\,\mathrm{m}$ height. Then, the gain in impact leads to a false identification of source regions and the $300\,\mathrm{m}$ assumption overestimates their impact.

Overall, the neglected dilution is identified as the major effect and the $300\,\mathrm{m}$ height assumption overestimates the impact of emission sources by $3-7\,\%$ in average.

So far, we discuss errors in the emission sensitivity that quantifies the relative impact of emission sources. However, the absolute impact of emission sources additionally depends on the source strength. The combination of emission sensitivity and emission flux density quantifies the contribution of upwind sources to the measured concentration at the receptor site. Hence, the errors amplify when strong emissions locally coincide with strong differences in emission sensitivity. This effect is confirmed in a typical application example of LPDMs (e.g., Forster et al., 2001; Duck et al., 2007; Stohl et al., 2007), that




simulates the CO source contributions from forest fires. Here, the $300\,\mathrm{m}$ ML height assumption overestimates the pick up of CO emissions by $16\,\%$ on average and up to $20\,\%$ at single time steps.

Furthermore, we also discuss the assumption of a $100\,\mathrm{m}$ ML height. This shallow ML height is widely-used in applications of FLEXPART (e.g., Stohl et al., 2007; Pan et al., 2014), since it represents the PBL height minimum that is used in the model. Then, a false identification of trajectories above the PBL height is excluded. However, the neglected dilution in this shallow ML height amplifies the overestimation of source contribution and leads to a distinct negative bias.

From this study, it can be concluded that the assumption of the ML height is of vital importance for the relationship between emission sources and the atmospheric composition at an observation site. Errors in ML height assumptions influence the emission sensitivity by the competing effects of loss / gain in concentration and gain / loss in impact. The weight of each effect depends on the ML height and the vertical distribution of air tracer particles. This is determined by the spatio-temporal variability in ML heights and the local meteorological conditions. They control the balance of both effects and finally the sensitivity to emission sources. The simplified assumption of a constant ML height close to the PBL height minimum neglects this spatio-temporal variability. Although the counteracting effects buffer these errors to some extent, the neglected dilution of emission fluxes introduces a major error in the quantification of emission sources. This affects two different applications:

When source contributions from upwind emission sources to an observation point are quantified, their impact is overestimated. When the emission fluxes of sources is estimated that matches the measured concentration at the downwind observation point, the source strength is underestimated. These errors may be incorporated into emission inventories. On a longer term, a dynamic ML height parametrization derived from local varying PBL heights should be included to LPDM models used to analyse the relation between sources of emission flux and receptor concentrations.

*Author contributions.* I. Hüser conducted the FLEXPART simulations, data analysis and wrote the manuscript. H. Harder co-organized the CYPHEX campaign and supervised I. Hüser. A. Heil extracted data from ECMWF and co-wrote the manuscript. J. Kaiser provided CO emissions from GFASv1.2 and co-wrote the manuscript.

*Acknowledgements.* ANY IDEAS who needs to be in here ???



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





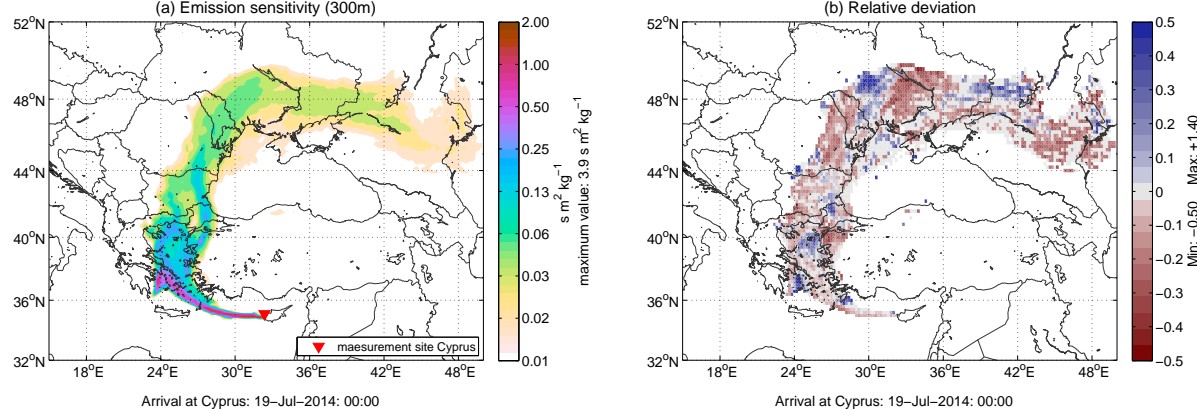

**Figure 1.** (a) Time integrated emission sensitivity referring to constant $300\,\mathrm{m}$ mixing layer assumption for a 5 day backward simulation started on 19th July 2014 00 UTC at Cyprus. Sensitivities below $0.01\,\mathrm{s\,m^{-2}\,kg^{-1}}$ are not considered. (b) Relative difference in emission sensitivity introduced by differences between local boundary layer heights and the constant $300\,\mathrm{m}$ mixing layer. Only absolute sensitivity differences above $0.0025\,\mathrm{s\,m^{-2}\,kg^{-1}}$ are used to calculate relative differences.

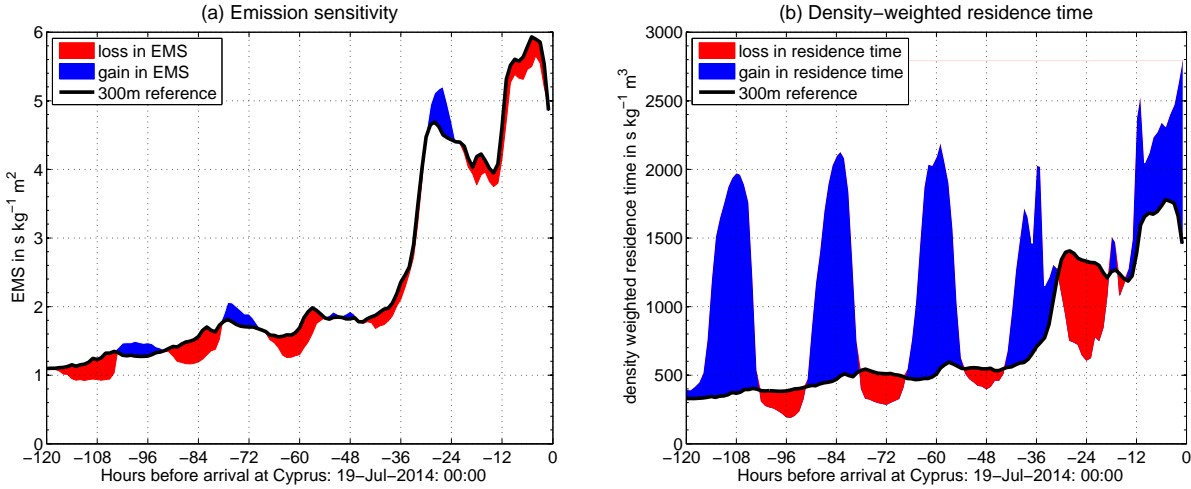

**Figure 2.** (a) Time profile of spatial integrated emission sensitivity (EMS) based on variable boundary layer heights with positive differences (blue areas) and negative (red areas) in reference to the $300\,\mathrm{m}$ assumption (black line). (b) Spatial integrated density-weighted residence time of boundary layer heights with differences shown by filled areas in respect to the $300\,\mathrm{m}$ layer.





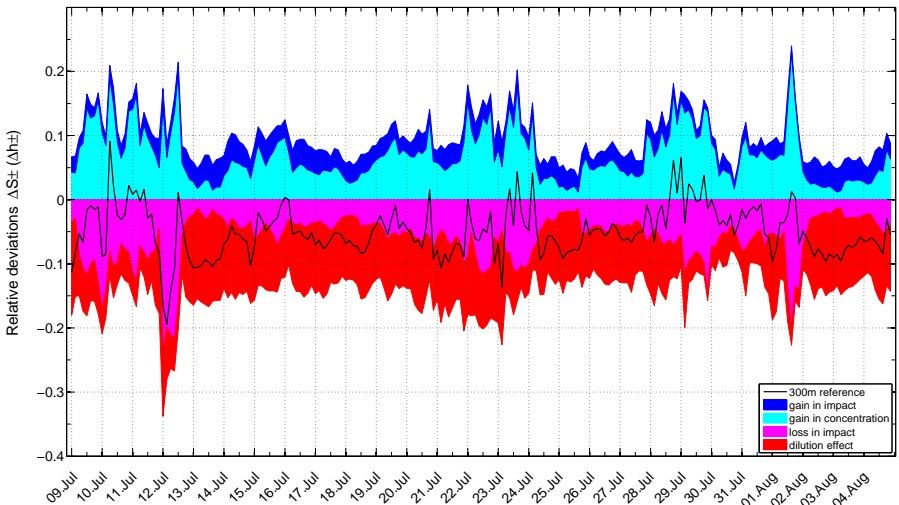

**Figure 3.** Temporal changes in the overall difference in emission sensitivity $\Delta S_{tot}$ enveloped by the positive and negative contributions that correspond to increase / decrease in mixing layer height $\Delta h\pm$ and the defined effects of eq.11 based on 216 simulations. Differences by mixing layer variations are calculated relatively in reference to the constant $300\,\mathrm{m}$ assumption.

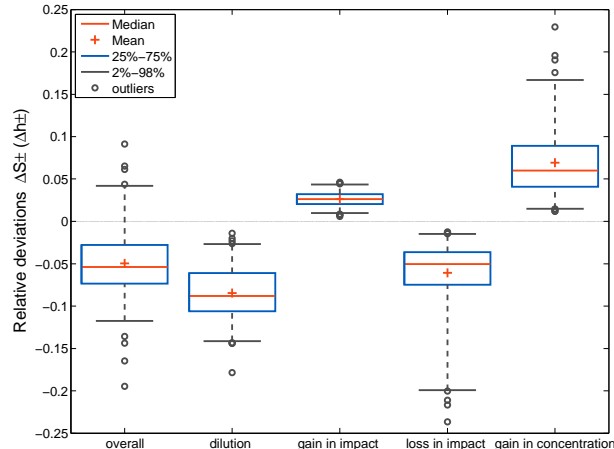

**Figure 4.** Box whisker plot for the overall differences in emission sensitivity and the contributions of the different effects in reference to the $300\,\mathrm{m}$ assumption based on 216 simulation. The whisker ends are defined by the 2nd and 98th percentile and remaining outliers considered by outlying circles.





**Figure 5.** (a) CO emissions for forest fires in Greece as a monthly mean for August 2007 provided by GFASv1.2 (Kaiser et al., 2012) and extrapolated on a grid of $0.2°$. (b) Zoom in the region of Greek forest fires for the time-integrated emission sensitivity of a 5 day backward simulation from 30th July 2014 15:00 UTC started at Cyprus and based on the $300\,\mathrm{m}$ mixing layer. (c) Time-integrated potential CO source contributions of forest fires in mass mixing ratios within the $300\,\mathrm{m}$ layer and (d) absolute differences introduced by mixing layer variations.





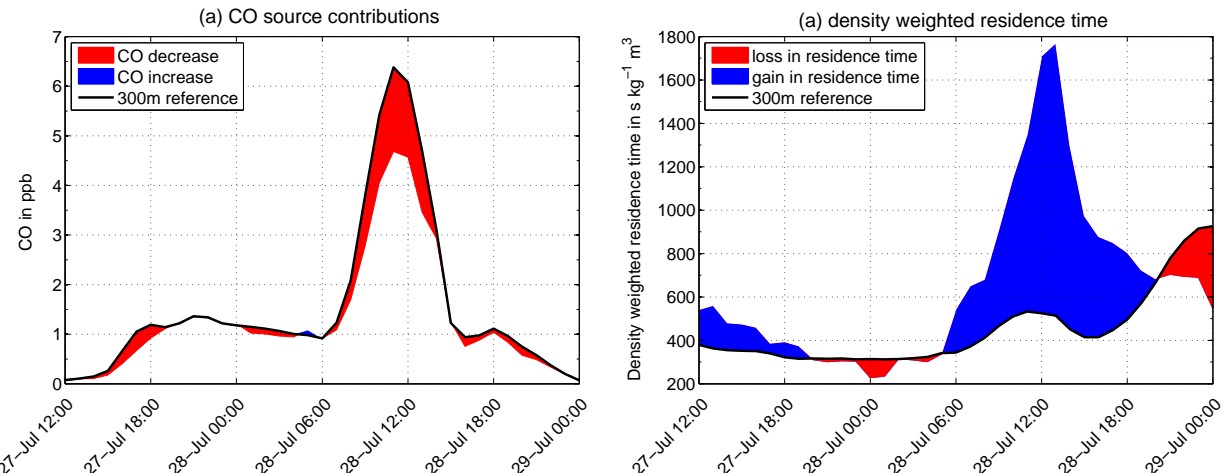

**Figure 6.** (a) Time profile of spatial integrated CO source contribution for the constant $300\,\mathrm{m}$ layer with differences introduced by mixing layer variations and (b) the corresponding profile of density-weighted residence time for the time window when Greek forest fires were passed. At 28-Jul 12:00 UTC a difference of $-26\,\%$ in CO contribution is accompanied by a mean boundary layer height of $1500\,\mathrm{m}$ and an increase of $226\,\%$ in residence time.