# Peer review of "Assumptions on footprint layer heights influence the quantification of emission sources: A case study for Cyprus"

_Atmospheric Chemistry and Physics, 2016_

## Referee Comment (RC1) · Anonymous Referee #1 · 10 Feb 2017

The paper "Assumptions on mixing heights influence the quantification of emission sources: A case study for Cyprus" by Hueser et al. describes the impact of two different assumptions on the height of the mixing layer in a lagrangian particle dispersion model which influence the emission sensitivity and hence the contribution of individual emission source locations to a local concentration enhancement.

The paper itself is clearly structured, however in some cases it is difficult to follow the conclusions as some important classification of individual cases are not explicitly explained in the manuscript.

A key issue for my criticism is a missing or only roughly touched differentiation between boundary layer height and mixing height, as these two quantities are only describing

the same effect for well mixed convective boundary layers. The manuscript makes no distinct differentiation for boundary layer stability conditions. Stable, e.g. nocturnal or boreal winter inversion boundary layers are characterised by much lower vertical mixing or even downward mixing and the surface emission sources are not even mixed throughout the much more shallow boundary layer. For neutrally stratified boundary layers (e.g. nocturnal residual layers) also no vertical mixing takes places. Hence mixing layers as described in the paper are only representative for a fraction of the occurrence in boundary layer types. A better characterisation for the types of boundary layer, as well as a PDF of boundary layer heights along the major emission sensitivity paths (e.g. emission sensitivity larger than 0.5 or even larger than 0.1 sm^2kg^-1) would help to understand the conditions for the scenarios.

Another key criticism the choice of using fire emissions for the case study together with the concept of the mixing layer. As fires represent local heat sources, the conditions of vertical motion and mixing of fire emissions are very different from other emission sources. This is one of the reasons why for fires usually effective emission heights are assumed in modelling, as fires trigger dry convection or often even pyroconvection which results in a much more enhanced vertical mixing and hence uplifting of air masses with enhanced biomass burning tracers. Therefore, the mixing layer height is a not well chosen concept for this emission type. However, for typical anthropogenic emissions from industry or road traffic, the concept is more suitable.

These two aspects should be properly discussed in a revised manuscript version. The statistical significance of the findings should be analysed for at least the climatological values.

Specific comments: Page 1, Line 19: "Local air composition is determined by transport processes....." This statement might be correct for locations which are far from emission sources. However, as a general statement this sentence is not correct, as local emissions and chemical production or destruction can be equally dominating the local air composition.

Page 2, Line 5: "...of the dispersion of an air mass by turbulence and convection in the lower troposphere." As these processes cannot directly resolved by lagrangian models on scales larger than a few kilometres, a good representation of these processes cannot be achieved. Statistical fluctuations are used as a tool to capture the main effects of these processes, only.

Chapter 2: The derivation of the equations is straight forward, such that they could be moved to an appendix, with a much more shortened explanation of the terms which are displayed in the figures (e.g. EQ2 and 4 are so similar except for the summation boundary, EQ3 and 5 except for the index in the denominator. Only EQ10 and EQ11 are of importance for the further analysis and discussion.

Page 7, line 30: What is the variability in PBL height? Here the PDF mentioned above would be a good way to illustrate this variability. Mean values usually do not help since they represent both daily and nocturnal conditions. However, as FLEXPART already provides the PBLH as output on the points of trajectory location, this information should be straight-forward to provide.

Page 8, line 5: This alternating pattern does not show a behaviour as expected from the change in diurnal and nocturnal boundary layer, which would result in larger areas with equal sign. Is this a consequence of the time integration? Are the differences statistically significant compared to the temporal internal variability?

Page 8: Line 15 to 20: Why is in the last 24 hours the difference only in negative direction and does not exhibit a gain in emission sensitivity any more? Comparing to Fig.6a the last 24 hours correspond to an increase in CO gain over Greece, however, this is not visible in Fig.2. Is this a consequence of different air masses and transport from a multitude of directions. If this is the case, then potentially a filtering of the data with respect to different wind directions or a cluster analysis of the origin of air masses is required a obtain a consistent picture.

Tab.1: Here a distinction in PBLH > MLH and PBLH < MLH height would be helpful.

Fig.3 and page 9 line 19-25: If the black line depicts the reference, it should be on the 0 line,as the difference of the reference to the reference is supposed to be zero. I would expect the black line to be the cumulative effect of all four processes, instead.

Technical points:

Page 8, line 20: time profile -> time series Page 9, line 32: Sentence structure appears wrong (verb is missing?). Page 11, line 24: ...both effects counteract with each other -> counteract each other Acknowledgements: This looks like a leftover from a preliminary manuscript version.

Section on data availability is missing.

---

## Referee Comment (RC2) · Anonymous Referee #2 · 20 Mar 2017

The paper "Assumptions on mixing heights influence the quantification of emission sources: A case study for Cyprus" intends to quantify the influence of varying boundary layer heights on the quantification of source contributions based on backward simulations performed with a Lagrangian Particle diffusion model (LPDM). In reality, however, this paper rather describes the influence of varying emission heights on the quantification of source contributions.

Looking at surface emissions (emission height <= 100 m), it is incorrect to assume that such emissions are taken up by particles that are within the mixing or boundary layer height. In the Lagrangian conceptual framework, such emissions are only taken up by the particles crossing this grid cell within the lowest model level, typically set to

0-100 m. Applications mentioned by the authors (like Stohl et al., 2007) are neither a simplification, nor do they assume constant mixing layer heights. In forward like in backward mode, the mixing layer dynamics is simulated in the model and thus does not need to be considered while counting the particles.

The framework, however, is different in case that the emission height exceeds the height of the lowest (surface) model layer, and especially in circumstances where emission heights are highly variable. This is especially the case for forest fire emissions, strongly depending on the dynamics of the fires and the temperatures prevailing in the plumes. Here, the variable emission heights need to be considered by including particles from higher levels in the quantification of source contributions. In this case, the effects described by the authors are relevant.

Therefore, these aspects should be properly discussed in a revised manuscript version, before the paper can be accepted.

---

## Author Response (AR1)

**Response to Reviewer comments**

We would like to thank both reviewers for their time, comments and suggestions for improvements. Basically, the two reviews are different in main aspects.

We think that both reviewers are right in the way they understand the paper.

Reviewer 1 agrees with our concept but criticises our application of fire emissions that is not suited for our concept.

In contrast, Reviewer 2 points out that the concept is not completely correct within the Lagrangian conceptual framework. However, for the specific case of fire emissions, Reviewer 2 raised the relevance of study.

Our new manuscript version is thoroughly revised. We adjusted our concept and think this new interpretation is in agreement with the Lagrangian framework. Furthermore, we replaced our application example of fire emissions and use effective emission injection heights as recommended by Reviewer 1. The adjusted concept is suited for this application, as raised by Reviewer 2, and the results better emphasise the relevance of the study.

Our response to the specific points are given below after the reviewer's remark in Italics. We start with Reviewer 2 as these comments introduced major revisions.

**Reviewer comments 2**

**General comments**

2.1 The paper "Assumptions on mixing heights influence the quantification of emission sources: A case study for Cyprus" intends to quantify the influence of varying boundary layer heights on the quantification of source contributions based on backward simulations performed with a Lagrangian Particle diffusion model (LPDM). In reality, however, this paper rather describes the influence of varying emission heights on the quantification of source contributions.

First of all, this comments points out that we mixed up the terms 'emission height' and 'mixing layer height'. Our use of the term 'mixing layer height' is misleading or even wrong as it describes the wellmixed layer of the PBL height (see also comment 2.3). The emission height defines the height where emission sources release pollutants to the atmosphere. The vertical mixing is then realised by turbulent processes up to the PBL height. This mixing layer dynamics is represented by the underlying meteorological model (more details in comment 2.2. and 2.4).

In case of fire emissions, this mixing layer dynamics is underrepresented as the fire-induced convection results in enhanced vertical mixing (details in comment 2.5 and 2.6).

Therefore, the reviewer is correct that we actually analysed the impact of varying emission heights for such a case.

However, the suggested title does not appropriately reflect our revised manuscript. We prefer to use the more technical term 'footprint layer' (FL) height instead of emission height which is widely used in the literature (e.g. Stohl et al., 2007a). This term offers more flexibility to describe any layer, in which emitted pollutants are assumed to affect tracer particles.

In our revised results, we consider two different cases:

In Section 3.1, we use the FL height to simulate the vertical mixing layer dynamics in the PBL height. This tests whether the tracer particles represent a well-mixed boundary layer (see also comment 2.4). In Section 3.2, we use the FL height to simulate the vertical mixing due to pyroconvection (see comment 2.5 and 2.6).

We adjusted our title using the new term footprint layer (FL) height and replaced the term 'mixing layer' with this new term throughout the whole manuscript.

Most relevant changes:

- $\rightarrow$  new title
- $\rightarrow$  new abstract
- $\rightarrow$  use of footprint layer (FL) height instead of mixing layer (ML) height
- → revised listing of methods in chapter: Methods 2.2

2.2 Looking at surface emissions (emission height <= 100 m), it is incorrect to assume that such emissions are taken up by particles that are within the mixing or boundary layer height. In the Lagrangian conceptual framework, such emissions are only taken up by the particles crossing this grid cell within the lowest model level, typically set to 0-100m.

We agree and have this made clear in the text, see comment 2.4.

2.3 Applications mentioned by the authors (like Stohl et al., 2007) are neither a simplification, nor do they assume constant mixing layer heights.

Yes, applications of FLEXPART do not use constant mixing layers, they use constant emission respective footprint layer heights. Our previous manuscript version uses the term in a misleading or

even wrong manner. In the revised version, the term 'mixing layer' refers to the well-mixed layer of the atmosphere, such as realised in (convective) PBL heights.

2.4 In forward like in backward mode, the mixing layer dynamics is simulated in the model and thus does not need to be considered while counting the particles.

This is correct as already specified in 2.2. The tracer particles should be evenly distributed throughout a well-mixed PBL height, representing the vertical mixing. Within the PBL, the pick up of pollutants is therefore independent of the FL height assumption. It should be possible to assume any FL height up the PBL. Typically, the lowest model layer, 0-100m, is recommended as minimum FL height in LPDM applications. This height includes the emission height of surface sources and corresponds to the minimum PBL height that guarantees a well-mixed layer and a reasonable particle counting statistics. Therefore, the reviewer is right that considering particles above the lowest model layer is actually not necessary. In our analysis in Section 3.1, we test extreme cases of extending the FL to the entire mixing layer, represented by the PBL, by dynamically using the local PBL height as FL height.

We have implemented this interpretation in the introduction and also adapted our conclusions. Now, our findings show that a FL height following local PBL heights yields systematic differences, with daytime and night-time sensitivity differences. At daytime when a well-mixed convective PBL can be assumed, the differences indicate that residual inaccuracies occur in the representation of the mixing layer dynamics.

→ Introduction: page 2, line 23 – page 3, line 7

 $\rightarrow$  revised abstract

ightarrow revised conclusions: page 13, lines 14-30

2.5 The framework, however, is different in case that the emission height exceeds the height of the lowest (surface) model layer, and especially in circumstances where emission heights are highly variable. This is especially the case for forest fire emissions, strongly depending on the dynamics of the fires and the temperatures prevailing in the plumes.

We are pleased that the reviewer supports the relevance of our study for the application on fire emissions.

Fires have their own dynamics and the fire-induced convection controls the vertical mixing of smoke constituents up to the injection height, which can exceed the PBL height and even reach the lower stratosphere. The mixing layer simulated in FLEXPART cannot account for this additional vertical transport. Thus using constant emission heights of 100m is only justified for surface emissions.  $\rightarrow$  Introduction: page 3, lines 7-11

2.6 Here, the variable emission heights need to be considered by including particles from higher levels in the quantification of source contributions. In this case, the effects described by the authors are relevant.

This comment encouraged us to recalculate our application example with emission heights from the plume rise model (PRM) injection heights of GFAS (Remy et al., 2017).

We combine a 5 days' period during the strong fire event in August 2007 in Greece with one of our case studies that shows transport routes passing through this fire region. For this case study, the fire plume heights even exceed local PBL heights and represent pyroconvection. We find that a 300m FL height assumption overestimates the CO source contributions by 60%. We also calculated the source contributions for the constant 100m FL assumption and the for the PBL heights. The 100m FL height

assumption introduces an overestimation of 77% while the difference for PBL heights is more moderate with 30%. These relative and absolute differences are summarised in the new Table 3. We also revised Fig.6 (now Fig.7). It shows the time series of CO source contributions and the density-weighted residence time for all four FL heights assumption: 100m, 300m, PBL height and fire plume height. The constant 100m and 300m FL heights show similar but still discernible results which confirms that the results are only weakly sensitive to small changes in FL heights. Thus the mixing layer dynamics may still be sufficiently represented by the trajectories for many applications. The PBL height assumption can reproduce the CO peak during daytime when convective PBL heights are similar to fire plume heights. However, this PBL height assumption erroneously elevates the background for stable shallow PBL heights at night.

This new analysis even shows a strong overestimation of fire emission impact for the use of constant FL heights up to 300m. Here, the PBL height might even be a reasonable assumption for biomass burning wherever observation-based fire plume heights are not available. We think that these findings are an additional value and emphasize the relevance of the study.

- ightarrow modifications for use of fire plume heights in chapter: Methods 2.2 and 2.4
- $\rightarrow$  completely revised chapter: Results 3.2
- $\rightarrow$  new Table 3
- $\rightarrow$  revised Fig. 6 (former Fig. 5)
- $\rightarrow$  revised Fig. 7 (former Fig.6)
- $\rightarrow$  revised conclusions: page 13, line 31 end
- ightarrow revised abstract

2.7 Therefore, these aspects should be properly discussed in a revised manuscript version, before the paper can be accepted.

We have introduced new simulations and adapted the text throughout the manuscript accordingly, see answers above.

**Reviewer comments 1**

**General comments**

1.1 The paper "Assumptions on mixing heights influence the quantification of emission sources: A case study for Cyprus" by Hueser et al. describes the impact of two different assumptions on the height of the mixing layer in a Lagrangian particle dispersion model which influence the emission sensitivity and hence the contribution of individual emission source locations to a local concentration enhancement. The paper itself is clearly structured, however in some cases it is difficult to follow the conclusions as some important classification of individual cases are not explicitly explained in the manuscript.

We are pleased that the reviewer noted a clear structure in our manuscript. The revision based on Review 2 (see above) introduced major changes and new conclusions. We hope this new manuscript version is easier to follow.

1.2 A key issue for my criticism is a missing or only roughly touched differentiation between boundary layer height and mixing height, as these two quantities are only describing the same effect for well mixed convective boundary layers.

This comment ties into Comments 2.1.-2.4. We have now updated the manuscript to make clear that all mixing except pyroconvection is represented in the trajectories. As explained above, we focus on studying the numerical method of using a footprint layer (FL) to (a) test the representation of mixing in the case of a well-mixed boundary layer and (b) represent the pyrogenic mixing up to fire plume heights. Thus we now differentiate between mixing height, boundary layer height and footprint layer height, and only assume that the boundary layer height equals the mixing height in the case of a convective boundary layer.

However, within the Lagrangian framework, the tracer particles are effectively mixed within the simulated PBL height (Stohl et al. (2005), Section 3 second last paragraph), even for stable conditions. (see also comment 1.3)

 $\rightarrow$  new title

 $\rightarrow$  use of footprint layer (FL) height instead of mixing layer (ML) height throughout the manuscript

1.3 The manuscript makes no distinct differentiation for boundary layer stability conditions. Stable, e.g. nocturnal or boreal winter inversion boundary layers are characterised by much lower vertical mixing or even downward mixing and the surface emission sources are not even mixed throughout the much more shallow boundary layer. For neutrally stratified boundary layers (e.g. nocturnal residual layers) also no vertical mixing takes places. Hence mixing layers as described in the paper are only representative for a fraction of the occurrence in boundary layer types. A better characterisation for the types of boundary layer, as well as a PDF of boundary layer heights along the major emission sensitivity paths (e.g. emission sensitivity larger than 0.5 or even larger than 0.1 sm2kg--1) would help to understand the conditions for the scenarios.

We agree with the reviewer concerning the conditions for vertical mixing for different PBL boundary layer stability conditions. Unfortunately, as already mentioned in comment 1.2, LPDMs evenly distribute the tracer particles within the PBL. This trajectory distribution represents the vertical mixing independently of the stability conditions. Also, a minimum PBL height of 100m is assumed

that guarantees for minimum vertical mixing in stable conditions. Within the Lagrangian framework a discussion about vertical mixing in different PBL heights is thus limited.

Since the mixing layer dynamics is simulated in the model, our study rather tests the representation of this vertical mixing for the specific case of a well-mixed PBL. Furthermore, we only analysed systematic differences linked to the diurnal cycle of PBL heights. The differentiation of PBL heights only specifies convective PBL types at daytime and more stable types at night. As suggested by the reviewer, we present the variability of PBL heights for our single case study in a frequency distribution in the new Fig.1. 55% of PBL heights are above 300m and assumed to be represent convective PBL heights. 45% are below 300m and represent more stable conditions. We hope that our added figure about the frequency distribution supports the analysis of our case study in Section 3.1.1.

→ revisions in: Results 3.1.1, page 8, lines 4-13 → new Fig. 1

1.4: Another key criticism the choice of using fire emissions for the case study together with the concept of the mixing layer. As fires represent local heat sources, the conditions of vertical motion and mixing of fire emissions are very different from other emission sources. This is one of the reasons why for fires usually effective emission heights are assumed in modelling, as fires trigger dry convection or often even pyroconvection which results in a much more enhanced vertical mixing and hence uplifting of air masses with enhanced biomass burning tracers. Therefore, the mixing layer height is a not well chosen concept for this emission type.

We agree that that in the case of fire emissions the effective emission injection heights are better suited to represent the vertical distribution of pollutants. Especially, when the fire plume tops exceed the PBL height, the mixing layer dynamics simulated in LPDMs cannot account for the enhanced vertical mixing. Therefore, we recalculated the application example with fire plume top heights from the GFAS emission inventory. This suggestion is also in agreement to Reviewer 2 (see also comment 2.5 and 2.6). Our new analysis even shows a much stronger overestimation of emission impact for the use of constant FL heights up to 300m. These findings are an additional value to emphasize the relevance of the study.

ightarrow modifications for use of fire plume heights in chapter: Methods 2.2 and 2.4

- $\rightarrow$  revised section 3.2
- $\rightarrow$  revised conclusions: page 13, line 31 end
- ightarrow major changes in conclusions and abstract

**1.5: These two aspects should be properly discussed in a revised manuscript version**

We have differentiated PBL types in the manuscript text as far as this is compatible with the major revisions based on Review 2. We also follow the idea to use effective fire emission injection heights, recalculated our application example and adapted the conclusions accordingly.

**1.6 The statistical significance of the findings should be analysed for at least the climatological values.**

The study period of one month contains various weather and transport patterns which are not only typical for summertime conditions. This is now highlighted in Sect. 3.1.2, page 9 lines 26-32

**Specific comments**

Page 1, Line 19: "Local air composition is determined by transport processes....." This statement might be correct for locations which are far from emission sources. However, as a general statement this sentence is not correct, as local emissions and chemical production or destruction can be equally dominating the local air composition.

We should use the word 'co-determine' instead of 'determine'. Revised form: "Transport processes in the atmosphere co-determine local air composition as ..."  $\rightarrow$  page 1, line 21

Page 2, Line 5: "...of the dispersion of an air mass by turbulence and convection in the lower troposphere." As these processes cannot directly resolved by lagrangian models on scales larger than a few kilometres, a good representation of these processes cannot be achieved. Statistical fluctuations are used as a tool to capture the main effects of these processes, only.

This formulation is misleading and the content of the citation not presented correctly. Revised form: "Since turbulence is stochastic a large number of trajectories should correctly represent the dispersion of an air mass in the lower troposphere (Stohl et al., 2002)."  $\rightarrow$  page 2, lines 6-7

Chapter 2: The derivation of the equations is straight forward, such tha they could be moved to an appendix, with a much more shortened explanation of the terms which are displayed in the figures (e.g. EQ2 and 4 are so similar except for the summation boundary, EQ3 and 5 except for the index in the denominator. Only EQ10 and EQ11 are of importance for the further analysis and discussion.

We agree that the derivation is straight forward. However, we appreciate formulas to clarify our methods. Therefore, we just shorten the section containing formulas.

Page 7, line 30: What is the variability in PBL height? Here the PDF mentioned above would be a good way to illustrate this variability. Mean values usually do not help since they represent both daily and nocturnal conditions. However, as FLEXPART already provides the PBLH as output on the points of trajectory location, this information should be straight-forward to provide.

We expect a distinct variability for a case study with continental transport routes just because of a pronounced diurnal cycle in PBL heights over land surfaces. To clarify the variability, we added a plot of the frequency distribution for PBL heights during the 5 days' simulation (see also comment 1.2).  $\rightarrow$  page 8, lines 6-13

Page 8, line 5: This alternating pattern does not show a behaviour as expected from the change in diurnal and nocturnal boundary layer, which would result in larger areas with equal sign. Is this a consequence of the time integration? Are the differences statistically significant compared to the temporal internal variability?

This plot (new: Fig.2) is basically used as an introduction to the analysis. The time-integrated emission sensitivity is usually the standard application to localise and quantify emission sources.

Therefore, we use this application to show the differences in sensitivity when varying FL heights are used. This point is better emphasis ed in the revised version.

Furthermore, the time-integration sums up positive and negative sensitivity differences at single time steps, i.e. Fig 2b only shows the temporal net-effect. However, this spatial pattern has still a link to the diurnal cycle of emission heights. The trajectories spread over time and, therefore, pass distinct regions preferred over day or night. This is actually the background to further analyse the impact of temporal variations in PBL heights in the next paragraph.

→ page 8, lines 14-30

Page 8: Line 15 to 20: Why is in the last 24 hours the difference only in negative direction and does not exhibit a gain in emission sensitivity any more? Comparing to Fig.6a the last 24 hours correspond to an increase in CO gain over Greece, however, this is not visible in Fig.2.

The gain in emission sensitivity in the last 12 hours results from a PBL height that exceeds the 300m layer. This is obvious in Fig2b (now 3b) that indicates a gain in residence time in the last 12 hours. The strong dilution in the extended PBL height finally causes the loss in emission sensitivity. Actually, it's untypical that the PBL height exceeds the 300m layer in the last 6 hours (21-3 local time). Since the measurement site is located on a hilly ground at a distance of 10km to the coast, PBL heights are calculated in an inhomogeneous environment on a 0.2° grid. Thus, they suffer stronger uncertainties than above a homogenous surface. During the last hours, the trajectories get denser sharing the same grid cell and the calculated PBL heights are more sensitive to these uncertainties. Therefore, the effects during the last 12 hour, especially 6 hours, can exhibit an unexpected behaviour.

Fig. 6 (now Fig. 7) is completely new as we used different CO emission fluxes and fire plume heights. The time series shows a time window in the middle of the simulation period and is thus not comparable to Fig.2a (now Fig. 3a).

**Tab.1: Here a distinction in PBLH > MLH and PBLH < MLH height would be helpful.**

In the revised version, we do no longer use the term mixing layer height and refer to the term footprint layer height. We think, that it is not reasonable to adapt Table 1. This suggestion would be only valid for this specific case for a comparison to PBL heights. This table should also be applicable in a more general sense, for example, the comparison between constant FL heights and variable fire plume heights. Thus, the table generally lists the four possible effects of FL height changes.

Fig.3 and page 9 line 19-25: If the black line depicts the reference, it should be on the 0 line, as the difference of the reference to the reference is supposed to be zero. I would expect the black line to be the cumulative effect of all four processes, instead.

Thanks to the careful review. There is an error in the legend. The black line is the cumulative effect and not the 300m reference. The relative differences are calculated in reference to the emission sensitivity of the 300m layer. Thus, the reference is zero.

**Technical points:**

Page 8, line 20: time profile -> time series

This is modified in the revised version.

Page 9, line 32: Sentence structure appears wrong (verb is missing?).

A comma and a verb improves this sentence: "When PBL heights fall below 300m, the concentration of emitted mass flux is intensified and less trajectories are captured by the shallower layer."

Page 11, line 24: ...both effects counteract with each other -> counteract each other

This is modified in the revised version.

Acknowledgements: This looks like a leftover from a preliminary manuscript version.

This is updated.

Section on data availability is missing.

A note about data availability is added at the end of the conclusions.

[revised manuscript text omitted]

10 modifications to calculate the sensitivity for a dynamical FLML height. Furthermore, we develop a method to compare the effects of different FLML height assumptions. In Section 3.1 we compare the emission sensitivity for a constant FLML height of 300 m and a dynamical height derived from local PBL heights. In an application example in Section 3.2, we simulate the CO contributions that result from hypothetical forest fires. Here, we replace the FL heights with the altitude of plume top and quantify the differences in source contributions. Finally, in Section 4 we summarise and discuss the results.

**15 2 Method**

**2.1 FLEXPART simulations**

We have performed backward simulations with the LPDM model FLEXPART 9.2 (Stohl et al., 2005) during the CYPHEX campaign. The FLEXPART model simulations have been driven by operational meteorological input data of the European Centre for Medium-Range Weather Forecasts (ECMWF) with  $0.2^{\circ} \times 0.2^{\circ}$  spatial resolution (derived from T799 spectral trun-

- 20 cation). The model domain extends from 20°W to 70°E and 20°N to 70°N covering Europe, northern Africa and western Asia. To provide a temporal resolution of 1 hour a combination of analyses at 00, 06, 12 and 18 UTC and short forecasts data at intermediate time steps was used. Backward simulations were started from the measurement point in the Northwest of Cyprus by releasing 10000 neutral inert air tracer particles during an 1 hour time interval which are followed as trajectories over 5 days.
- The positions of the tracer particles are calculated on a 3-d grid with a horizontal resolution of  $0.2^{\circ}$  that corresponds to the resolution of the input data on the model domain. A vertical resolution of 58 layers is used extending from 100 m (minimum height of the planetary boundary layer (PBL) set in the model) to an altitude of 10000 m. The height of the layers gradually increases from 20 m to 1000 m. The temporal resolution corresponds to the input data and is available in hourly time steps within the simulation period of 120 h. In total, 216 simulations covering 9th July 2014 00 UTC to 4th August 2014 21 UTC in
- 30 3 hour time steps were carried out. They cover the entire period of the measurement campaign.

The output of FLEXPART is a 4-d function of emission sensitivity (Stohl et al., 2007a), three space dimension plus time, derived from the positions of all trajectories. It describes the relation between any emission source that is passed by the trajectories and the concentration of the respective atmospheric substance at the receptor. Since atmospheric emissions are

diluted by mixing in adjacent air, emission sources are specified as emitted mass per time and volume in units of kg m-3 s-1, denoted by q. Receptor concentrations are expressed in a conservative quantity as mass mixing ratios, specified by  $\chi$  (unitless). Then, the emission sensitivity is determined by the sum of the residence time T of all trajectories in this adjacent volume of air divided by the local air density  $\rho$  in kg m-3 without transmission correction (Seibert and Frank, 2004)

$$\quad \frac{\partial \chi}{\partial q_{ijkn}} = \frac{T_{ijkn}}{\rho_{ijkn}}.$$
(1)

It is computed in the predefined 3-d output grid that uses the indices i, j, k to specify the spatial position  $x_i, y_j, z_k$  in the centre of each grid box. The fourth index n determines the time step  $t_n$  within the simulation period. This emission sensitivity can be interpreted as a source-weight factor. It describes the source contribution to the mass mixing ratio at the receptor relative to the source strength when any transport losses are ignored.

10

In addition to the described standard output, FLEXPART provides the PBL height for all individual tracer particles at their geographical position at each time step. The PBL heights are calculated by the model according to Vogelezang and Holtslag (1996) and use the concept of the critical Richardson number that is based on surface sensible heat fluxes and surface stresses available from the ECMWF input data (Section 3 in Stohl et al., 2005).

**2.2 Surface Emission Sensitivity**

- 15 Since we are interested in area sources at ground level, the emission sensitivity for volume sources in Eq. 1 is slightly modified to derive the sensitivity to surface emissions. The emission flux of an area source  $q_A$  in units of kg s-1 m-2 is diluted over a vertical layer adjacent to the ground, the footprint layer (FL)mixing layer (ML) of height h, following  $q = q_A h^{-1}$ . With this assumption, all grid boxes of height  $z_k \le h_{ijn}$  are attributed to the mixing volume in the FL height and influenced by pollutants. Here we use a variable FL height  $h_{ijn}$  in space and time that is specified by the corresponding indices i, j, n. Then,
- 20 the emission sensitivity S to surface area sources in units of  $s m^2 kg^{-1}$  is obtained with the density-weighted residence time  $\hat{T}_{ijn}$  in this mixing volume as

$$\widehat{T}_{ijn} = \sum_{k=1}^{z_k \le h_{ijn}} \frac{T_{ijkn}}{\rho_{ijkn}}$$

$$S_{ijn} = \frac{\widehat{T}_{ijn}}{h_{ijn}}.$$
(2)
(3)

In this study, we use different FLML heights and have implemented threetwo methods:

25

- The first method uses the assumption of a constant FLML height within the PBL height (Seibert and Frank, 2004). Here, we choose a layer of h = 300 m which is used in published studiesthat is close to the typically expected minimum of PBL heights. Additionally, this height is slightly above the PBL minimum in the model of 100 m. This allows to analyse the case that the assumed FL height exceeds the PBL height. Then, all grid boxes with height  $z_k \le h$  are attributed to the mixing volume. The emission sensitivity S to surface area sources is obtained from eq. 1 with the density-weighted

residence time  $\widehat{T}$  in this mixing volume as

$$\widehat{T}_{ijn} = \sum_{k=1}^{z_k \leq \underline{h}} \frac{T_{ijkn}}{\rho_{ijkn}}$$

$$S_{ijn} = \frac{\widehat{T}_{ijn}}{\underline{h}}.$$
(2\*)
(3\*)

The emission sensitivity S in units of  $s m^2 kg$  is proportional to the density-weighted residence time of all trajectories within the ML height. Even for spatial or temporal integration, the emission sensitivity is independent of the ML height.

 The second method is new in this study and uses a FL height hijn that is variable in space and time.dynamic ML height. This ML height hijn is variable in space and time and specified by the corresponding indices i, j, n. Then, eq. 2\* and 3\* are replaced by

$$\widehat{\mathcal{I}}_{jn} = \sum_{k=1}^{\frac{z_k \le hijn}{p_{ijkn}}} \frac{T_{ijkn}}{p_{ijkn}} \tag{4*}$$

$$S_{ijn} = \frac{\widehat{T}_{ijn}}{h_{ijn}}.$$
(5\*)

15

10

5

With this dynamic FLML assumption, the emission sensitivity depends on local varying FLML heights. It is hence not proportional to the density-weighted residence time, when spatially or temporal integrated. To implement this dynamic ML height, we We use local PBL heights calculated by the FLEXPART model to implement this dynamic FL height. However, from the output of FLEXPART only PBL heights along the particles' trajectories are available. To obtain gridded PBL heights from all single particle positions, we adapt the methodology described in Stohl et al. (2005), Section 8.1section 11.1. Their eq. (55) is modified and calculates We modify their Eq. (50) and calculate the PBL height h on the spatio-temporal output grid

$$h = \sum_{p=1}^{N} (f_p h_p)$$
(4)

20

25

with N being the total number of particles,  $h_p$  the PBL height of particle p and  $f_p$  the fraction of the particle attributed to the respective grid cell. To calculate this fraction, we use a uniform kernel with bandwidths of  $0.2^{\circ}$  that corresponds to the output grid.

The third method is specifically developed for the application on fire emissions. It is based on the the plume rise model (PRM) injection heights from the Global Fire Assimilation System (GFAS) (Rémy et al., 2017; Paugam et al., 2015). These fire observation-based plume heights are used as a variable FL height to simulate the emission impact of fire emissions provided by the GFAS (Kaiser et al., 2012).

6

**2.3 Analysing the impact of footprint layer height assumptions**

To analyse the impact of footprint layer (FL)<del>ML</del> height variations, we calculate the emission sensitivity for both constant and dynamic FL<del>ML</del> heights and analyse their differences. The absolute difference is described by

$$\Delta S_{ijn} = S_{ijn}(h_{ijn}) - S_{ijn}(h). \tag{5}$$

5 Here,  $\Delta S_{ijn}$  is defined such that it is positive when FLML height differences introduce a higher emission sensitivity compared to the constant 300 m FLML height assumption. Differences in FLML height are determined in the same way,  $\Delta h_{ijn} = h_{ijn} - h$ , with positive sign for an increase.

To analyse the changes in emission sensitivity that are introduced by varying FLML heights, Eq.  $35^{**}$  is differentiated with respect to h,

$$\quad \frac{dS(h)}{dh} = \frac{d}{dh}\frac{\widehat{T}(h)}{h} = \frac{1}{h}\frac{d\widehat{T}}{dh} - \frac{1}{h^2}\widehat{T}.$$
(8\*)

Rearranging the equation to

$$\frac{dS(h)}{dh} = \frac{\widehat{T}}{h} \frac{1}{dh} \left( \frac{d\widehat{T}}{\widehat{T}} - \frac{dh}{h} \right).$$
(6)

and discretising Discretising the differentials results in the simple relation

$$\frac{\Delta S}{S} = \frac{\Delta \hat{T}}{\hat{T}} - \frac{\Delta h}{h}.$$
(7)

- 15 The relative change in density-weighted residence time  $\left|\frac{\Delta \hat{T}}{\hat{T}}\right|$  describes the gain / loss in impact since a deeper FLML height can capture more trajectories. The second term describes the relative change in FLML height  $\left|\frac{\Delta h}{h}\right|$ . It quantifies the dilution of emitted substances and represents the gain / loss in concentration. Thus, to describe the relative change in emission sensitivity, we need to analyse the overall difference of both effects. Additionally, both effects are coupled and interact with each other. An increase in FLML height results in an increase in residence time and is accompanied by a stronger dilution. Hence, the changes
- 20 in emission sensitivity are the result of the counteracting effects: gain in impact and loss in concentration (dilution effect). It is crucialCrucial is which of both turns out to be the dominating effect. This interaction is described in a case-by-case analysis presented in Tab. 1 respectively for increasing and decreasing FLML heights, respectively.

Following this case-by-case analysis, the overall difference in emission sensitivity, spatially and / or temporally integrated, is the results of these four characteristic differences

25
$$\Delta S_{tot} = \Delta S^{-}(\Delta h^{+}) + \Delta S^{+}(\Delta h^{+}) + \Delta S^{+}(\Delta h^{-}) + \Delta S^{-}(\Delta h^{-}).$$
 (8)
overall difference dilution effect <0 gain in impact >0 gain in concentration >0 loss in impact <0

[revised manuscript text omitted]
. HereIn fact, the 'fixed' FLML height applicationassumption capitalises on over- and underestimationsthese local maxima and minima compensating for each other. Therefore, a systematic trend based on any spatial surface properties is not given. The remaining net effect reveals an alternating spatial pattern. This is linked to the diurnal cycle of the PBL height since the trajectories pass distinct regions preferred at day or night-time.

Therefore, we<del>To further</del> explore the impact of temporal variations in FLML height respective PBL heights and <del>, the emission</del> 35 sensitivity is spatially integrated 
[revised manuscript text omitted]
             | $745\mathrm{ppb}$                                              | +77~%                                                                                     |
| $300\mathrm{m}$   | $676\mathrm{ppb}$                                              | +60%                                                                                      |
| PBL               | $555\mathrm{ppb}$                                              | +32%                                                                                      |
| fire plume height | $420\mathrm{ppb}$                                              | -                                                                                         |
|                   | footprint height
100 m
300 m
PBL
fire plume height | footprint heightCO contribution100 m745 ppb300 m676 ppbPBL555 ppbfire plume height420 ppb |

[revised manuscript text omitted]